# Stunting Status of Ever-Married Adolescent Mothers and Its Association with Childhood Stunting with a Comparison by Geographical Region in Bangladesh

**DOI:** 10.3390/ijerph19116748

**Published:** 2022-05-31

**Authors:** Md. Ahshanul Haque, Barbie Zaman Wahid, Md. Tariqujjaman, Mansura Khanam, Fahmida Dil Farzana, Mohammad Ali, Farina Naz, Kazi Istiaque Sanin, ASG Faruque, Tahmeed Ahmed

**Affiliations:** Nutrition and Clinical Services Division, icddr,b, Dhaka 1212, Bangladesh; barbiezwahid@gmail.com (B.Z.W.); md.tariqujjaman@icddrb.org (M.T.); mansura@icddrb.org (M.K.); fahmidaf@icddrb.org (F.D.F.); md.ali@icddrb.org (M.A.); farina.naz@icddrb.org (F.N.); sanin@icddrb.org (K.I.S.); gfaruque@icddrb.org (A.F.); tahmeed@icddrb.org (T.A.)

**Keywords:** stunting, adolescent, association, Bangladesh

## Abstract

The adolescence period is considered a life stage worthy of strategic health investments since it is a critical period of physical and neuro-maturational development. Adolescent girls face different health difficulties in that phase of life. Children born to adolescent mothers are at a higher risk of undernutrition. This paper aims to estimate the prevalence of stunting among adolescent mothers and their children in Bangladesh by time period and determine the associated factors of adolescent maternal stunting status. We also sought to establish the relationship between maternal and childhood stunting by comparing the geographical regions in Bangladesh. We derived data from the nationally representative Bangladesh Demographic and Health Survey, which was conducted between 2007 and 2017/18. The outcome variables of this study were ever-married adolescent girls’ stunting status and their children’s stunting status. Interaction analysis between administrative division and maternal stunting status was conducted with childhood stunting as the outcome variable to investigate the impact of maternal stunting status on their children’s stunting compared to geographical location. Our results indicated that in comparison to other divisions, the frequency of stunting among children and adolescent mothers was higher in the Sylhet region. It also revealed that children whose mothers were stunted had a 2.36 times increased chance of being stunted. Our study suggests that education for women could help them attain self-sufficiency and, as a result, reduce the prevalence of poor childhood nutrition, especially stunting.

## 1. Introduction

Globally, over 3 billion people, or nearly half of the world’s population, are under 25 years of age [1]. Among them, roughly 1.2 billion are adolescents, aged between 10 and 19 years [2]. The adolescence period is considered a life stage worthy of strategic health investments since it is a critical period of physical and neuro-maturational development [3]. Adolescent girls face different health difficulties in that phase of life. Among these difficulties, teenage pregnancy is the most serious because of its negative effects on both the mother and the child’s health. It is estimated that about 16 million adolescent girls give birth every year; 95% of those births take place in low- and middle-income countries. Adolescent pregnancy frequently leads to school dropout, which has a negative impact on young women’s education and income. The nutritional status of women is similarly impacted [3]. Moreover, early marriage is associated with negative psychological and socioeconomic implications for both mothers and their children [2].

Children born to adolescent mothers are at a higher risk of undernutrition [4]. Furthermore, adolescents’ underweight has a significant effect on their child’s nutritional status. Children of adolescent mothers are eight times more likely to be stunted than children of older mothers [3]. The mother’s body mass index (BMI) and age at marriage, the child’s size at delivery, and antenatal clinic visits all have a strong independent relationship with the child’s nutritional status. Moreover, several factors including poor maternal nutritional status, low level of maternal education, limited access to health care services, poor complementary feeding practices, poor living conditions, and adolescent pregnancy are associated with child undernutrition [3]. More so, newborns of adolescent mothers are twice as likely to suffer from stillbirth, low birthweight, and neonatal death compared to newborns of mothers aged 20–29 years [5,6].

Malnourished girls are more likely to have short stature in adulthood, perinatal mortality, and prematurity. Low-birthweight newborns were shown to be quite common among mothers with heights less than 140 cm. Furthermore, a WHO collaborative research of maternal anthropometry and pregnancy outcomes suggests that maternal height and weight should be used for screening in its service application [7]. In prior investigations, there was a substantial correlation between maternal age and the occurrence of stunting. Hence, intergenerational health correlations between a mother and her offspring have been assessed using maternal height as a marker. Maternal stature itself especially is found to be associated with the child’s nutritional status. Studies reported that children born to mothers with short stature were more likely to be stunted. This relationship indicates that women stunted during their own childhood are more likely to have stunted children [8,9]. Previous research found various adverse health outcomes, including nutritional outcomes, in children of low-height mothers (of short stature). As a result, the height of the mother can be used to predict the nutritional status of the child. Furthermore, children of short-stature mothers (height of 145 cm or less) are more likely to be stunted and underweight [10]. Bangladesh has made significant progress in reducing the rate of maternal and childhood stunting in all divisions of Bangladesh. In Sylhet, however, no such decline has been documented, while all the divisions have reported reduced stunting to varying degrees [11].

To estimate chronic malnutrition for adolescent mothers, their age, BMI, and height may not be applicable. In this situation, age-specific indicators based on the WHO 2007 reference for children aged 5–19 years are appropriate to calculate adolescent mothers’ chronic malnutrition [12]. To the best of our knowledge, no investigation has been conducted in Bangladesh to estimate the prevalence and determine the trends and associated factors of adolescent mothers’ stunting with their child’s stunting status using nationally representative data. Furthermore, there was a scarcity of research addressing the regional comparative study of stunting among adolescent mothers and their children’s stunting status in Bangladesh. Hence, the first objective of this study was to determine the associated factors of adolescent mothers’ stunting status. The second objective was to assess the survey round-specific association between an adolescent mother’s stunting and her child’s stunting status. Finally, the third objective was to assess the association between childhood stunting and the adolescent mother’s stunting by geographical region.

## 2. Materials and Methods

### 2.1. Data Sources

In this study, we derived data from the nationally representative Bangladesh Demographic and Health Survey (BDHS), which was conducted between 2007 and 2017/18. We used the “women individual recode (BDIR)” dataset which included data of ever-married women aged 15–19 years. The detailed survey design was described elsewhere [11]. The BDHS survey is a two-stage stratified sampling design. The stratification was performed by separating each division into urban and rural areas. In the BDHS 2007 survey, Bangladesh was divided into six divisions: Barisal, Chittagong, Dhaka, Khulna, Rajshahi, and Sylhet. However, from BDHS 2011 to 2014, the Rajshahi division was divided into two divisions, Rajshahi and Rangpur. Furthermore, in 2017–2018, the Dhaka division was divided into two divisions, Dhaka and Mymensingh. Since we appended four rounds of datasets from 2007, 2011, 2014, and 2017–2018, we used six divisions and merged the divided division with the previous division (Appendix A). The final analysis included 1348, 2004, 2023, and 1951 ever-married adolescents from BDHS 2007, 2011, 2014 and 2017/18, respectively. We restricted our analysis to the ever-married adolescent group rather than the whole age group since the adolescence period is the last stage for intervention to increase the height, which ultimately plays a significant role in reducing the stunting status. Furthermore, there is a lack of measurement tools for calculating the stunting status of adults (age ≥ 20 years). Therefore, in this study, we focused on ever-married adolescent women.

### 2.2. Variables under Study

The outcome variables of this study were ever-married adolescent girls’ stunting status and their children’s stunting status. Ever-married adolescent girl’s stunting status was defined as the height-for-age z-score (zhfa) < −2, where the indicator, zhfa, was calculated using the WHO Child Growth Standards 2007 STATA macro package, who2007.ado [12]. Another outcome variable in our study was childhood stunting, which was measured based on the height/length-for-age z-score (HAZ/LAZ) < −2. Z-scores were calculated through the use of the “2006 WHO standards for children where z-score scale = (observed value—average value of the reference population)/standard deviation value of reference population” [13,14]. To begin with, a list of several other variables was finalized through the results obtained from descriptive and bivariate analyses as well as the literature review. Household characteristics were as follows: division as the geographical area, place of residence (urban or rural), wealth index, number of household members, improved toilet, source of drinking water, and religion. Ever-married adolescent girls’ characteristics were as follows: height (cm), weight (kg), age (years), height-for-age z-score, stunting status, maternal education, husband’s education, husband’s age, woman’s own health care, making of major household purchases, visits to her family or relatives, all of the three decisions, attitudes to domestic violence, at least four ANC visits from a medically trained provider, use of contraception, delivery type. Child’s characteristics included child’s sex, child’s age (months), and childhood stunting. The description of these variables was given in the published report [11].

### 2.3. Statistical Analysis

Statistical software Stata (release 14; StataCorp LP, College Station, TX, USA) was used to analyze the data. Bar diagrams were used to visualize the outcome variables. Frequency and proportion for qualitative variables and mean and standard deviation for quantitative variables were used to summarize the data. The outcome variables as well as all other variables were segregated across different BDHS rounds. To see the relationship between the outcome variables and all the independent variables, the chi-squared test was used for qualitative variables and the t-test was used for quantitative variables. Due to binary outcomes, logistic regression was used to assess the factors associated with the outcome variables. First, simple logistic regression analysis was used to examine the bivariate association. Then, multiple logistic regression was used to calculate the adjusted odds ratios (aORs) as the strength of association between the outcome variables and the relevant independent variables. The variables were included in multiple regression models based on our bivariate findings as well as the literature review.

To assess the survey round-specific association between the adolescent mother’s stunting and the child’s stunting status, we conducted multiple logistic regression analyses for each round as well as for the pooled data where the outcome variable was the childhood stunting status and the exposure variable was the adolescent mother’s stunting status after adjusting the relevant covariates such as geographical area, place of residence, wealth index, number of HH members, religion, age (years), maternal education, husband’s education, husband’s age, attitudes to domestic violence, use of contraception, delivery type, working status, child’s age, child’s sex, and BDHS round for the overall model.

To assess the association of the maternal stunting status with the child’s stunting with a comparison by geographical region, interaction analysis between the administrative division and the maternal stunting status was analyzed where the outcome variable was the childhood stunting. During the comparison of the findings between the geographical areas, the Sylhet region was defined as a reference group in the model to compare with the rest of the geographic areas since the prevalence of maternal and child malnutrition was higher in this region. The strength of association was presented as aORs with respective 95% confidence intervals. Furthermore, for the explanation of the regression model, *p*-value < 0.05 was considered to indicate significance. Since the data were nationally representative, the sampling weight was considered in each model during data analysis.

## 3. Results

### 3.1. General Characteristics

The general characteristics of households, ever-married adolescent girls with their children are given in Table 1.

### 3.2. Prevalence of Stunting

This paper, with the nationally representative BDHS 2007 to 2017/18 data, provided detailed geographical level stunting prevalence between adolescent mothers and their children. The prevalence of adolescent girls’ stunting was 43.9%, 39.8%, 38.6%, and 38.5% in the 2007, 2011, 2014, 2017/18 surveys, respectively. On the other hand, the prevalence of stunting among their children was 36.0%, 40.1%, 35.1%, and 32.1% in the 2007, 2011, 2014, and 2017/18 surveys, respectively. Comparing these statuses between the geographical regions, we found that the status in the Sylhet region was worse than in any other regions (Figure 1 and Figure 2).

### 3.3. Factors Associated with Adolescent Stunting

Table 2 represents the unadjusted and adjusted odds ratios which describe the factors associated with adolescent stunting. The value of pseudo R^2^ was 0.0329. The chi-squared value for the Hosmer–Lemeshow goodness-of-fit test was 6955 with a *p*-value = 0.423 which indicated that the model was well-fit. The model’s sensitivity and specificity were 21% and 87%, respectively. The correct classification rate was 61% and the area under the receiver operating characteristic (ROC) curve was 0.618. According to the post-estimation findings as well as the value of the area under the curve of the ROC, the predictive performance was low. Then, we may have to figure out other indicators which were not included in the model but might have an influence on childhood stunting.

The odds of having stunted adolescent mothers in the geographical regions such as Barisal (OR: 0.76 (95% CI: 0.60, 0.98); *p*-value = 0.032), Chittagong (OR: 0.78 (95% CI: 0.62, 0.98); *p*-value = 0.031), Dhaka (OR: 0.92 (95% CI: 0.73, 1.17); *p*-value = 0.508), Khulna (OR: 0.61 (95% CI: 0.48, 0.78); *p*-value < 0.001), and Rajshahi (OR: 0.78 (95% CI: 0.62, 0.97); *p*-value = 0.028) were rather different compared to another region, Sylhet. The urban adolescents were more stunted (OR: 1.24 (95% CI: 1.08, 1.43); *p*-value = 0.003) than the rural ones. According to the socioeconomic status, the adolescents from the poorest households (OR: 1.65 (95% CI: 1.30, 2.10); *p*-value < 0.001) were more likely to be stunted than the richest households. Similarly, adolescents from the poorer (OR: 1.64 (95% CI: 1.33, 2.04); *p*-value < 0.001), middle (OR: 1.39 (95% CI: 1.13, 1.71); *p*-value = 0.002), richer (OR: 1.32 (95% CI: 1.09, 1.60); *p*-value = 0.005) households were also more likely to be stunted compared with adolescents from the richest households. Education level below secondary (OR: 1.31 (95% CI: 1.14, 1.51); *p*-value < 0.001), age (OR: 1.01 (95% CI: 1.01, 1.02); *p*-value < 0.001), working status (OR: 1.17 (95% CI: 0.99, 1.38); *p*-value = 0.058), husband’s age (OR: 0.96 (95% CI: 0.95, 0.98); *p*-value < 0.001), husband’s education level below secondary (OR: 1.25 (95% CI: 1.10, 1.42); *p*-value < 0.001), experience with domestic violence (OR: 1.16 (95% CI: 1.02, 1.32); *p*-value = 0.027), and Muslim religion (OR: 1.36 (95% CI: 1.10, 1.68); *p*-value = 0.005) were also significantly associated with adolescent stunting. If we see the trend, the status of adolescent stunting in BDHS rounds 2014 (OR: 0.79 (95% CI: 0.66, 0.94); *p*-value = 0.007) and 2017/18 (OR: 0.73 (95% CI: 0.62, 0.87); *p*-value < 0.001) was found less often than in round 2007.

### 3.4. Association between Maternal and Childhood Stunting

Figure 3 depicts the status of childhood stunting by maternal stunting stratified by category round of the BDHS survey. The overall prevalence of stunting was 45% among those children whose mothers were stunted, whereas the status was 26% among those children whose mothers were non-stunted.

We found that children of the stunted mothers were more stunted as well. Table 3 shows that the odds of having childhood stunting among the stunted adolescent mothers was 2.36 times higher (OR: 2.36 (95% CI: 1.96, 2.84); *p*-value < 0.001) compared to the non-stunted adolescent mothers using pooled data. The round-specific odds ratios as the strength of associations between the adolescent mother’s stunting and the child’s stunting status were as follows: OR: 2.62 (95% CI: 1.74, 3.92), *p*-value < 0.001 in 2007; OR: 1.66 (95% CI: 1.19, 2.32), *p*-value = 0.003 in 2011; OR: 2.36 (95% CI: 1.58, 3.53), *p*-value < 0.001 in 2014; and OR: 3.78 (95% CI: 2.59, 5.53), *p*-value < 0.001 in 2017–2018.

Controlling for the geographical region, there was no significant difference in childhood stunting among the regions for stunted adolescent mothers. On the other hand, the odds of having childhood stunting among non-stunted adolescent mothers in such regions as Barisal (OR: 0.43 (95% CI: 0.26, 0.70); *p*-value < 0.001), Chittagong (OR: 0.37 (95% CI: 0.23, 0.60); *p*-value < 0.001), Dhaka (OR: 0.37 (95% CI: 0.23, 0.59); *p*-value < 0.001), Khulna (OR: 0.32 (95% CI: 0.20, 0.52); *p*-value < 0.001), Rajshahi (OR: 0.39 (95% CI: 0.25, 0.60); *p*-value < 0.001), and Sylhet (OR: 0.83 (95% CI: 0.47, 1.45); *p*-value = 0.509) were rather different compared to stunted mothers in Sylhet region. However, in the Sylhet region, the adolescent mother’s stunting was not associated with the childhood stunting (Table 4).

## 4. Discussion

Adolescence is a time of significant physiological, sexual, neurological, and behavioral changes; adolescence growth surge necessitates more energy and nutrients. Furthermore, the proximity of adolescents to biological maturity and adulthood may provide final chances to avoid adult health problems. Therefore, in order to monitor the health of a population or a community, an assessment of the growth and nutritional status of children and adolescents is required.

The adolescent mothers with primary school education were more likely to be stunted than those with secondary school education. Likewise, previous studies revealed an association between the BMI-for-age z-score and educational status. The highest grade completed was positively associated with the height-for-age z-score < −2 in our study. A study previously revealed that adolescents who obtain even a basic education are more aware than those who receive no education of how to use available resources to enhance their own and their families’ nutritional health. Education may assist one to make autonomous decisions, get acceptance from other members at home, and gain more access to household resources that are critical to one’s nutritional condition [15]. More so, a well-educated woman is more likely to send all of her children to school, breaking the cycle of ignorance; she is also more likely to employ childhood survival techniques including appropriate breastfeeding, immunization, oral rehydration therapy, and family planning. As a result, educating women could help to reduce the prevalence of poor childhood nutrition, particularly stunting [16]. In addition, the current study found that adolescent employment is associated with their stunting condition. According to a previous study, employment was positively associated with the BMI-for-age z-score and the height-for-age z-score. Unemployed adolescents had a much higher chance of being underweight and stunted than employed adolescents [15]. Employed adolescents have the opportunity to contribute to the total family income and some degree of financial independence. Furthermore, it can be interpreted that this financial autonomy facilitates them to make decisions to improve their overall nutrition and health. Stunting was much lower in Sylhet among the mothers with a higher education, among the poorest quintile, and in non-Muslim families. Other important initiatives to reduce stunting, in addition to improving economic conditions, include improving health and education levels, particularly in Sylhet.

Our study has also demonstrated that adolescents from poorer households were more likely to be stunted when compared to those from affluent households. According to the findings from a previous study, children and adolescents from the poorest households are approximately two times more likely to be stunted when compared to those from the richest households. As a result, the prevalence of stunting is likely to be higher among the poorest children and adolescents than among the richest. Food-insecure households had a low monthly income. Food security is associated with family’s economic status which is eventually associated with the stunting status of children [17]. Hence, socioeconomic status could also be related to the stunting status of adolescents, which conforms to our findings. Our findings also indicate that the prevalence of stunting was higher in the urban area compared to the rural area. This result disagrees with a couple of studies conducted in Yemen and Ethiopia. According to a study carried out in Ethiopia, rural adolescent women are more likely to suffer from chronic energy deficiency than adolescent women in urban areas [15]. This can be explained by the fact that in rural areas, poorer education, poor socioeconomic status, lack of potable water, infectious illness prevalence, and nutritional awareness are more prevalent than in urban areas [18]. Childhood undernutrition comparisons between urban and rural populations imply that urban people are better off than rural populations. However, these comparisons may obscure the significant socioeconomic disparities that occur in urban regions. Data from the Demographic and Health Surveys for 11 countries in three regions were used to test the hypothesis that intraurban differences in child stunting were larger than intrarural differences and that stunting was equally prevalent among the urban and rural poor. Using principal components analysis, a socioeconomic status index based on household assets, housing quality, and service availability was produced separately for rural and urban parts of each country. Stunting in the poorest urban quintile was practically on par with that of poor rural inhabitants in most countries, which supports our findings [19].

We found that the proportion of childhood stunting was higher in Sylhet division. Previously, a study [18] conducted in Bangladesh revealed that the prevalence of stunting was greater among children from Sylhet division when compared to the other regions, which is also aligned with our findings. According to the BDHS survey, Sylhet division is regarded as a low-performing region [20]. Despite the fact that Sylhet division receives a big amount of foreign income, residents have been seen investing more in land purchases and homes than in health, education, and food consumption, which could explain such findings [18]. Children from the poorest families were more likely to be stunted, and the likelihood of being stunted decreases as the wealth index rises. This could be owing to the fact that lower-income people are vulnerable to buying less nutritious food and may be forced to consume an inadequate amount of nutrients, less than the minimal daily requirement [18].

In this study, we observed that children of the stunted mothers appeared to be stunted as well. A previous multilevel analysis conducted in Nigeria revealed that in under-five children, the mother’s BMI was found to be substantially related with stunting and severe stunting. That study also depicted that mothers with a BMI of less than 18.5 kg/m^2^ were significantly more likely than mothers with a BMI of 25 kg/m^2^ or higher to have stunted children [21]. In addition to this, a study found a 216 g difference in birthweight between women of height of 143 cm or less and women of height of 162 cm or more. Another study found that birthweight increased with the increase in maternal height [7]. Furthermore, previous studies have also found that maternal BMI is a significant risk factor for poor intrauterine growth and low birthweight, both of which are known drivers of stunting and severe stunting in early childhood [21]. Thus far, these findings are in line with our study. Furthermore, in a comparative cross-sectional study conducted in Bangladesh, the mother’s BMI, which is a measure of her nutritional status, was found to be substantially linked to severe and moderate stunting of the children. The impact of the mother’s nutritional state starts in utero and continues for at least the first 6 months after birth when the infant is completely reliant on the mother for all of its nutritional needs. Additionally, low birthweight and low BMI of mothers were found to be important predictors of stunting in children in a cross-sectional study conducted in Tanzania. Another study, also conducted in Tanzania, reported that, compared to children born to mothers with a higher BMI, those born to mothers with a lower BMI were more likely to be severely stunted [22].

Maternal height is an important predictor of childhood stunting. In order to improve maternal nutrition, pertinent policy level interventions should be conducted to provide them with adequate food to improve their nutritional status at adolescence as children born to malnourished mothers are at a higher risk of malnutrition and stunting. Additionally, a previous literature analysis found that the mother’s education remains a powerful indicator of the child’s nutritional status. Childhood stunting is significantly related to the mothers’ education level, and the odds of childhood stunting are higher for mothers with no education or lower than secondary education relative to mothers that have at least secondary education. Hence, appropriate measures should be taken at the policy level to put emphasis on girl–child education. As a result of their improved educational background, rates of maternal employment may also increase. On the other hand, a prior literature review indicated that maternal experience of domestic violence plays a significant role in compromising child health by impairing child nutrition and growth. Measures should be emphasized to strengthen efforts to protect women from domestic violence, with a big picture perspective to minimize childhood stunting. Furthermore, larger-scale programs to promote maternal and child health should be undertaken among the vulnerable populations in the underdeveloped countries.

## 5. Strengths and Limitations

There are several strengths of this study. It is based on data from nationally representative demographic and health surveys that used internationally validated questionnaires and applied nutritional status definitions combined with a strong methodology. It had a high response rate (>94%) indicating the generalizability of the research findings to the entire nation. The current study provides a comparison of the 2007, 2011, 2014, and 2017/18 surveys and examines the changes in the determinants of stunting in Bangladesh which have not been reported before. This comparison gives an indication for future interventions and provides a benchmark for future comparisons [23]. There are some limitations of the study. Firstly, the data used in this study were cross-sectional, and they were unable to establish a causal relationship. Moreover, looking into the study sampling distribution across eight regions over four periods, it reveals that sampling distribution varies across the regions. Thus, adjusted weighted factors may explain some differences in stunting reductions, especially for Dhaka where covariates play a minimum role in explaining the reduced level of stunting [20]. Furthermore, with the rapid rise of urbanization in Bangladesh, the definition of urban and rural areas has shifted. As a result, certain places that were previously classified as rural in prior BDHSs were classified as urban in more recent BDHSs, potentially leading to urban–rural calculation discrepancies.

## 6. Conclusions

Our study suggests that educating women could help them attain self-sufficiency and, as a result, reduce the prevalence of poor childhood nutrition, especially stunting. Comprehensive initiatives encouraging maternal education and nutrition as well as income-generating activities should be implemented nationwide to enhance childhood nutritional status. In addition, strategies to address domestic violence against women should be implemented to help reduce childhood stunting in a developing country.

## Figures and Tables

**Figure 1 ijerph-19-06748-f001:**
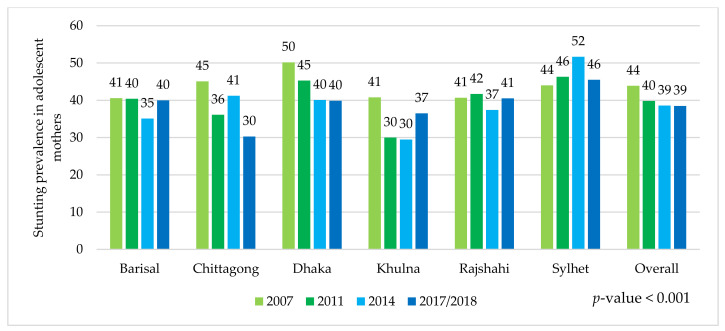
Stunting prevalence in adolescent mothers stratified by geographical region and round of the BDHS survey.

**Figure 2 ijerph-19-06748-f002:**
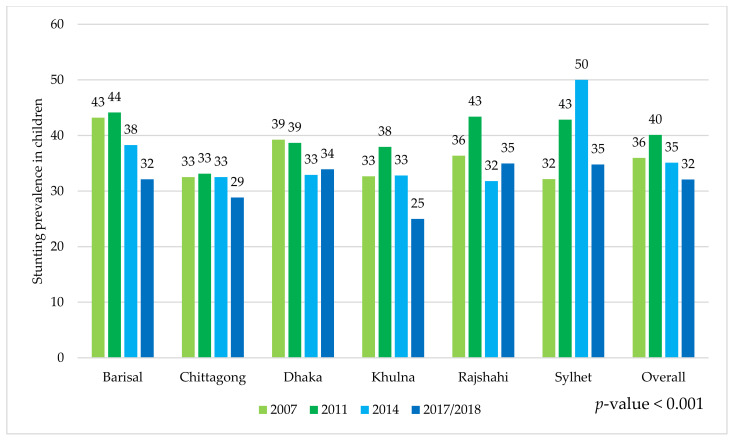
Stunting prevalence in children whose mothers were adolescents was stratified by geographical region and round of the BDHS survey.

**Figure 3 ijerph-19-06748-f003:**
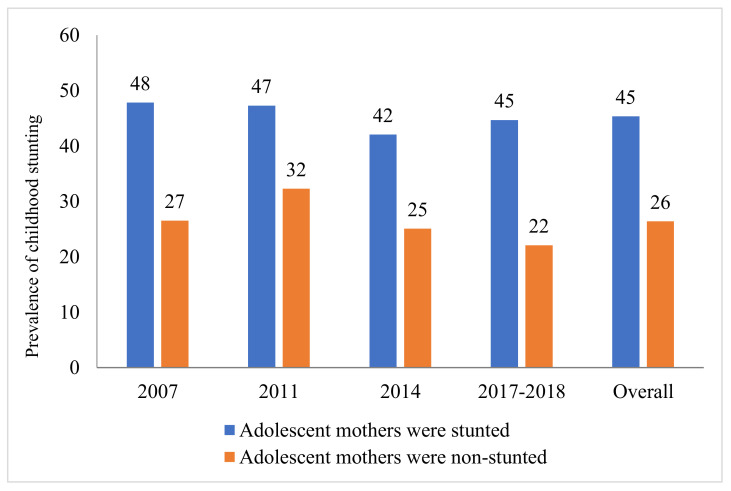
Status of childhood stunting by maternal stunting stratified by category round of the BDHS survey.

**Table 1 ijerph-19-06748-t001:** General characteristics of the participants stratified by round of the BDHS survey.

**Indicators, % (n)**	**BDHS Round**			
	**2007**	**2011**	**2014**	**2017–2018**
**Household characteristics**	**N = 1348**	**N = 2004**	**N = 2023**	**N = 1951**
Geographical area				
Barisal	13.0 (175)	12.3 (247)	13.8 (279)	11.4 (222)
Chittagong	18.3 (247)	15.6 (313)	17.6 (356)	15.7 (307)
Dhaka ^!^	21.1 (284)	17.8 (356)	16.8 (339)	26.9 (525)
Khulna	15.5 (209)	15.7 (315)	13.6 (276)	13.3 (259)
Rajshahi ^$^	20.7 (279)	30.5 (611)	27.9 (565)	25.1 (490)
Sylhet	11.4 (154)	8.1 (162)	10.3 (208)	7.6 (148)
Place of residence				
Urban	30.4 (410)	30.5 (611)	31.1 (629)	31.4 (612)
Rural	69.6 (938)	69.5 (1393)	68.9 (1394)	68.6 (1339)
Wealth index				
Poorest	14.3 (193)	16.9 (338)	19.3 (391)	21.2 (413)
Poorer	21.7 (293)	24 (481)	20.5 (415)	22.4 (437)
Middle	22.6 (305)	22.2 (445)	22.6 (457)	21 (409)
Richer	21.2 (286)	22.2 (444)	22.6 (458)	20.7 (403)
Richest	20.1 (271)	14.8 (296)	14.9 (302)	14.8 (289)
Number of HH members ^1^	6.1 (3.3)	5.9 (2.9)	5.8 (2.8)	5.9 (2.7)
Improved toilet	35.1 (458)	45.7 (893)	59.4 (1185)	52.5 (976)
Source of drinking water	79 (1064)	81 (1623)	81 (1636)	77.3 (1499)
Religion				
Other	7.2 (97)	8.8 (177)	7.3 (147)	6.0 (116)
Muslim	92.8 (1251)	91.2 (1827)	92.7 (1876)	94.1 (1835)
**Ever-married adolescent girls’ characteristics**	N = 1328	N = 1961	N = 2006	N = 1917
Height (cm) ^1^	150.4 (6.2)	150.9 (5.4)	151.2 (5.6)	151.3 (5.3)
Weight (kg) ^1^	44.9 (7.4)	45.3 (7.3)	46.5 (8)	48 (8.2)
Age (years) ^1^	18 (1.3)	17.9 (1.5)	18 (1.3)	18.2 (1.2)
Height-for-age z-score ^1^	1.9 (0.9)	1.8 (0.8)	1.8 (0.8)	1.8 (0.8)
BMI-for-age z-score ^1^	0.5 (2.6)	0.6 (0.9)	0.4 (1)	0.2 (1.1)
Stunting	43.9 (583)	39.8 (781)	38.6 (774)	38.5 (739)
Maternal education				
No education	10.7 (144)	7.2 (145)	5.2 (105)	2.3 (45)
Primary	28.4 (383)	27.4 (550)	25.6 (517)	22.2 (434)
Secondary	57.8 (779)	58.5 (1172)	59.7 (1208)	61 (1191)
Higher	3.1 (42)	6.8 (137)	9.5 (193)	14.4 (281)
Husband’s education				
No education	24 (323)	16.7 (334)	13.9 (282)	11 (214)
Primary	34.5 (465)	33.7 (676)	33.1 (669)	34.4 (672)
Secondary	31.8 (429)	39.7 (795)	40.8 (825)	37.5 (731)
Higher	9.7 (131)	9.9 (199)	12.2 (247)	17.1 (334)
Husband’s age ^1^	26.6 (6)	26.2 (6.1)	26 (4.7)	25.9 (4.4)
Had the ability to take decisions herself (or jointly with her husband)				
i. Own health care	37.5 (506)	47.4 (925)	45.8 (907)	57.1 (1079)
ii. Major household purchases	32 (432)	39.2 (766)	37.4 (741)	43 (814)
iii. Visits to her family or relatives	34.1 (459)	43.7 (854)	40.7 (806)	49.5 (936)
All of the three decisions	18 (242)	28.8 (563)	25.9 (513)	32.7 (618)
Attitudes to domestic violence	64.5 (870)	67.2 (1346)	70.2 (1421)	80.5 (1570)
At least four ANC visits from a medically trained provider	8.6 (116)	11.4 (229)	10.7 (216)	20.1 (393)
Use of contraception	39.2 (529)	48.2 (966)	50.3 (1017)	48.9 (955)
Delivery type				
Caesarean section	7.5 (56)	12.4 (128)	21 (197)	28.9 (251)
Non-caesarean	92.5 (693)	87.6 (905)	79 (742)	71.1 (618)
**Child’s characteristics**	N = 753	N = 1042	N = 1061	N = 987
Child’s age (months)^1^	18.1 (13.5)	17.6 (14.3)	17.8 (13.7)	17.9 (14.1)
Child’s sex				
Male	49.8 (375)	52.1 (543)	51.0 (541)	52.6 (519)
Female	50.2 (378)	47.9 (499)	49.0 (520)	47.4 (468)
Childhood stunting	36.0 (240)	40.1 (366)	35.1 (332)	32.1 (293)

^!^ Mymensing division was united with Dhaka. ^$^ Rangpur division was united with Rajshahi. ^1^ Mean (SD).

**Table 2 ijerph-19-06748-t002:** Factors associated with the stunting status of ever-married adolescents.

Indicators		Unadjusted OR (95% CI)	*p*	Adjusted OR (95% CI)	*p*
Geographical area					
	Sylhet	Reference		Reference	
	Barisal	0.69 (0.55, 0.87)	0.002	0.76 (0.60, 0.98)	0.032
	Chittagong	0.64 (0.52, 0.80)	<0.001	0.78 (0.62, 0.98)	0.031
	Dhaka ^!^	0.84 (0.67, 1.04)	0.108	0.92 (0.73, 1.17)	0.508
	Khulna	0.54 (0.43, 0.68)	<0.001	0.61 (0.48, 0.78)	<0.001
	Rajshahi ^$^	0.75 (0.61, 0.92)	0.006	0.78 (0.62, 0.97)	0.028
Place of residence					
	Rural	Reference		Reference	
	Urban	1.08 (0.96, 1.23)	0.203	1.24 (1.08, 1.43)	0.003
Wealth index					
	Richest	Reference		Reference	
	Poorest	1.75 (1.43, 2.13)	<0.001	1.65 (1.30, 2.10)	<0.001
	Poorer	1.62 (1.35, 1.94)	<0.001	1.64 (1.33, 2.04)	<0.001
	Middle	1.23 (1.02, 1.49)	0.028	1.39 (1.13, 1.71)	0.002
	Richer	1.22 (1.01, 1.47)	0.036	1.32 (1.09, 1.60)	0.005
Education					
	At least secondary	Reference		Reference	
	Below secondary	1.68 (1.49, 1.89)	<0.001	1.31 (1.14, 1.51)	<0.001
Age (months)		1.01 (1.01, 1.01)	<0.001	1.01 (1.01, 1.02)	<0.001
Working status					
	Not working	Reference		Reference	
	Working	1.29 (1.12, 1.50)	0.001	1.17 (0.99, 1.38)	0.058
Husband’s age		0.96 (0.95, 0.98)	<0.001	0.96 (0.95, 0.98)	<0.001
Husband’s education					
	At least secondary	Reference		Reference	
	Below secondary	1.59 (1.42, 1.78)	<0.001	1.25 (1.10, 1.42)	0.001
Domestic violence					
	No	Reference		Reference	
	Yes	1.05 (0.93, 1.19)	0.411	1.16 (1.02, 1.32)	0.027
Religion					
	Other	Reference		Reference	
	Muslim	1.31 (1.07, 1.60)	0.010	1.36 (1.10, 1.68)	0.005
Round					
	2007	Reference		Reference	
	2011	0.86 (0.73, 1.01)	0.065	0.88 (0.74, 1.04)	0.128
	2014	0.79 (0.66, 0.93)	0.006	0.79 (0.66, 0.94)	0.007
	2017–2018	0.75 (0.64, 0.88)	0.001	0.73 (0.62, 0.87)	<0.001

^!^ Mymensing division was united with Dhaka. ^$^ Rangpur division was united with Rajshahi.

**Table 3 ijerph-19-06748-t003:** Survey round-specific association between the adolescent mother’s stunting and the child’s stunting status.

BDHS Round	Unadjusted OR (95% CI)	*p*	Adjusted * OR (95% CI)	*p*
2007	2.54 (1.75, 3.69)	<0.001	2.62 (1.74, 3.92)	<0.001
2011	1.88 (1.39, 2.55)	<0.001	1.66 (1.19, 2.32)	0.003
2014	2.17 (1.52, 3.10)	<0.001	2.36 (1.58, 3.53)	<0.001
2017–2018	2.85 (2.07, 3.91)	<0.001	3.78 (2.59, 5.53)	<0.001
Overall	2.31 (1.95, 2.73)	<0.001	2.36 (1.96, 2.84)	<0.001

* Adjusted odds ratio of having childhood stunting among the stunted mothers compared to the non-stunted ones was calculated using multiple logistic regression after adjusting for the relevant covariates such as geographical area, place of residence, wealth index, number of HH members, religion, age (years), maternal education, husband’s education, husband’s age, attitudes to domestic violence, use of contraception, delivery type, working status, child’s age, child’s sex, and BDHS round for the overall model. The outcome variable was the childhood stunting status and the exposure variable was the adolescent mother’s stunting status.

**Table 4 ijerph-19-06748-t004:** Association between the childhood stunting and the adolescent mother’s stunting by geographical region.

Indicators	Unadjusted OR (95% CI)	*p*	Adjusted * OR (95% CI)	*p*
**Interaction of adolescent mother’s stunting and geographical area**				
Stunted mother × Sylhet	Reference		Reference	
Stunted mother × Barisal	1.39 (0.86, 2.25)	0.173	1.13 (0.69, 1.84)	0.632
Stunted mother × Chittagong	0.76 (0.49, 1.20)	0.239	0.83 (0.52, 1.32)	0.435
Stunted mother × Dhaka ^!^	0.93 (0.60, 1.45)	0.742	0.89 (0.56, 1.42)	0.624
Stunted mother × Khulna	1.05 (0.64, 1.71)	0.847	1.03 (0.62, 1.73)	0.900
Stunted mother × Rajshahi ^$^	1.16 (0.76, 1.76)	0.491	0.92 (0.60, 1.43)	0.725
Non-stunted mother × Barisal	0.51 (0.32, 0.81)	0.004	0.43 (0.26, 0.70)	0.001
Non-stunted mother × Chittagong	0.37 (0.24, 0.57)	<0.001	0.37 (0.23, 0.60)	<0.001
Non-stunted mother × Dhaka ^!^	0.44 (0.28, 0.68)	<0.001	0.37 (0.23, 0.59)	<0.001
Non-stunted mother × Khulna	0.35 (0.22, 0.55)	<0.001	0.32 (0.20, 0.52)	<0.001
Non-stunted mother × Rajshahi ^$^	0.45 (0.29, 0.68)	<0.001	0.39 (0.25, 0.60)	<0.001
Non-stunted mother × Sylhet	0.73 (0.43, 1.24)	0.245	0.83 (0.47, 1.45)	0.509

^!^ Mymensing division was united with Dhaka. ^$^ Rangpur division was united with Rajshahi. * Adjusted for the place of residence, wealth index, number of HH members, religion, age (years), maternal education, husband’s education, husband’s age, attitudes to domestic violence, use of contraception, delivery type, working status, child’s age, child’s sex, and BDHS round.

## Data Availability

The data that support the findings of this study are available on request from the corresponding author. The data are not publicly available due to privacy or ethical restrictions.

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
