# Peer review of "Stunting Status of Ever-Married Adolescent Mothers and Its Association with Childhood Stunting with a Comparison by Geographical Region in Bangladesh"

_ijerph, 2022, doi:10.3390/ijerph19116748_

Round 1

Reviewer 1 Report

Dear Authors,

the reviewed article addresses several important issues such as the determinants of stunting in adolescent mothers, the relationship between maternal and child stunting, and the incidence of stunting over the years and its variation in different regions of Bangladesh.   
The authors' involvement in the research process should be appreciated; however, some doubts are raised about the presentation and interpretation of the results. All of them will be presented below with reference to particular parts of the text.

Introduction
The relationship between malnutrition and stunting, which undoubtedly exists, should be clarified, but it is necessary to confirm this by citing the results of studies confirming this fact.
It is advisable to clearly formulate both the main objective of the study and the sub-objectives (the results indicate that these were: to determine the determinants of stunting in adolescent mothers, to determine the relationship between stunting in mothers and children, and to determine the differences in the occurrence of the phenomenon under study in different regions of the country). 

Material and methods
This chapter lacks an explanation of why the study focused on ever-married adolescent mothers - what arguments supported this choice?

Results
The results are properly presented and described. 

Discussion 
In the discussion, the authors briefly presented the main results of the study, and then rather selectively (though extensively) compared them with the results of previous analyses. It seems that according to the idea of the presentation of the results it would be appropriate to first discuss the determinants of stunting in adolescent mothers (it is not clear why the authors focused only on education and wealth), then to reflect on the impact of maternal stunting on the occurrence of this trait in children, as well as the intensity of the analyzed phenomenon in the Sylhet region.     
It seems rather simplistic to associate stunting only with maternal education, income and malnutrition resulting from these determinants. The results obtained testify to a broader context of the phenomenon that can be suggested, although education seems to be crucial in modifying the impact of other determinants of the problem under study. 
A minor formal remark - it would be appropriate to exclude from the discussion the chapter: Strengths and weaknesses and future research (which was not mentioned). 

Conclusions
This chapter should focus on the main conclusions of the study - references to the situation in Bangladesh seem unnecessary. 

Author Response

Reviewer 1

The reviewed article addresses several important issues such as the determinants of stunting in adolescent mothers, the relationship between maternal and child stunting, and the incidence of stunting over the years and its variation in different regions of Bangladesh. The authors' involvement in the research process should be appreciated; however, some doubts are raised about the presentation and interpretation of the results. All of them will be presented below with reference to particular parts of the text.

Introduction

The relationship between malnutrition and stunting, which undoubtedly exists, should be clarified, but it is necessary to confirm this by citing the results of studies confirming this fact. It is advisable to clearly formulate both the main objective of the study and the sub-objectives (the results indicate that these were: to determine the determinants of stunting in adolescent mothers, to determine the relationship between stunting in mothers and children, and to determine the differences in the occurrence of the phenomenon under study in different regions of the country).

Response: We appreciate the reviewer’s insightful suggestion and have updated in accordance (page: 2; Line: 82-86).

Material and methods
This chapter lacks an explanation of why the study focused on ever-married adolescent mothers - what arguments supported this choice?

Response: Thank you for raising the issue. Now we have given the explanation in the method section (page: 3; Line: 102-107).

Results
The results are properly presented and described. 

Response: Thank you

Discussion: In the discussion, the authors briefly presented the main results of the study, and then rather selectively (though extensively) compared them with the results of previous analyses. It seems that according to the idea of the presentation of the results it would be appropriate to first discuss the determinants of stunting in adolescent mothers (it is not clear why the authors focused only on education and wealth), then to reflect on the impact of maternal stunting on the occurrence of this trait in children, as well as the intensity of the analyzed phenomenon in the Sylhet region. It seems rather simplistic to associate stunting only with maternal education, income and malnutrition resulting from these determinants. The results obtained testify to a broader context of the phenomenon that can be suggested, although education seems to be crucial in modifying the impact of other determinants of the problem under study.

Response: Thank you very much for the comments. We have reorganized the discussion section based on your suggestions (page: 11; Line: 321-331).

A minor formal remark - it would be appropriate to exclude from the discussion the chapter: Strengths and weaknesses and future research (which was not mentioned).

Response: Thank you very much for the comments. We have included a separate section based on your suggestions (page: 12; Line: 371).

Conclusions: This chapter should focus on the main conclusions of the study - references to the situation in Bangladesh seem unnecessary. 

Response: We thank the reviewer for pointing this out. We have revised accordingly (page: 12; Line: 390-397).

Reviewer 2 Report

Good work; nicely presented.

Author Response

Reviewer 2

Good work; nicely presented.

Response: Thank you very much for the comments.

Reviewer 3 Report

The work presented to me for review raises a very important problem. Early motherhood is a danger not only for the mother but also for the baby. Babies born to adolescent mothers are more likely to suffer from a range of illnesses, including stunted growth and the development of dwarfism. All activities aimed at reducing the percentage of teenage mothers deserve dissemination, therefore I believe that the work is ready for publication after a few corrections and additions.

-line 48- error in the word "mass"

-lines 45 and 50 are a repetition

-Please remove the gaps between the following parts of the introduction - they are unnecessary

-The authors compare the size of the problem in different regions of Bangladesh, but there are no characteristics of the entire country as well as individual regions. Please complete this, as this will increase the value of your work.

-I suggest moving some information from the introduction (e.g. lines 59-61) to the chapter: Discussion "

- Figures 1 and 2 no description of the y axis

- lines 229-245 are a description of the results, not a discussion

Author Response

Reviewer 3

The work presented to me for review raises a very important problem. Early motherhood is a danger not only for the mother but also for the baby. Babies born to adolescent mothers are more likely to suffer from a range of illnesses, including stunted growth and the development of dwarfism. All activities aimed at reducing the percentage of teenage mothers deserve dissemination, therefore I believe that the work is ready for publication after a few corrections and additions.

-line 48- error in the word "mass"

Response: Thank you. Now we have made necessary revisions (page: 2; Line: 47).

-lines 45 and 50 are a repetition

Response: Thank you. Now we have removed (page: 2; Line: 55-56).

-Please remove the gaps between the following parts of the introduction - they are unnecessary

Response: Thank you. Now we have removed the gap (whole text).

-The authors compare the size of the problem in different regions of Bangladesh, but there are no characteristics of the entire country as well as individual regions. Please complete this, as this will increase the value of your work.

Response: Thank you. Now we have re-analyzed (Supplementary Table 1)

-I suggest moving some information from the introduction (e.g. lines 59-61) to the chapter: Discussion "

Response: Thank you. Now we have made necessary revisions based on your suggestion (Page: 11; Line: 337-340).

- Figures 1 and 2 no description of the y axis

Response: We agree and have revised the text to address your cconcern  (Page: 6; Line: 177, 179).

- lines 229-245 are a description of the results, not a discussion

Response: Thank you. The text has been re-arranged in accordance to the reviewer (Result section) 

Reviewer 4 Report

Thank you for the opportunity to review the paper entitled: Stunting status of ever-married adolescent mothers and its association with childhood stunting with comparing the geographical region in Bangladesh.

In this work, the authors aimed to estimate the prevalence of stunting among adolescent mothers and their children in Bangladesh by time period, and determine the associated factors of adolescent maternal stunting status.

I have some comments regarding work.

1. The authors report the prevalence of stunting over four years. It might be interesting if the authors indicate if they found any effect related to the year in which the survey was carried out.

2. Could the authors explain why they used Sylhet as the geographic area to compare with the rest of the geographic areas.

3. Throughout the text, the authors report values ​​of p=0. This is an error that should be corrected; if the p-value is very close to 0, they should indicate p<0.001 since there are no p values ​​equal to 0.

4. For the estimation of the adjusted ORs, in the final step, the independent variables were added to the multiple regression model using the forward stepwise selection method. However, the forward stepwise selection method is used to select the most suitable model, so there may be variables that are not included in the final model. However, as shown by the authors, all the variables were selected despite not contributing to the model. This seems to be wrong. In addition, they do not indicate which were the criteria were used for the selection of these variables in the step-by-step selection method; that is, if the selection of the variables of the model it used the p values, the Akaike information criterion, the Bayesian information criterion or other criteria.

5. The values ​​of R2 for the models should be described.

6. In lines 162-164, the authors make the following statement: "these statuses by comparing with geographical regions, we found that the status in Sylhet region was worse than any other regions (Figure 1 and Figure 2)." However, in the text and in the figure, there is no statistical test or p-value to justify this statement.

7. Figure 3 is not the most suitable to represent OR values.

8. On lines 210-217, the authors show the results of the interactions. However, further explanation of the findings is required to give readers a better sense of the results.

Author Response

Reviewer 4

Thank you for the opportunity to review the paper entitled: Stunting status of ever-married adolescent mothers and its association with childhood stunting with comparing the geographical region in Bangladesh. In this work, the authors aimed to estimate the prevalence of stunting among adolescent mothers and their children in Bangladesh by time period, and determine the associated factors of adolescent maternal stunting status. I have some comments regarding work.

Response: Thank you very much for your comments. We have gone through your comment carefully to address them one by one. We hope the manuscript has been improved accordingly.

  1. The authors report the prevalence of stunting over four years. It might be interesting if the authors indicate if they found any effect related to the year in which the survey was carried out.

Response: Thank you. Now we have done additional analysis (Figure 3 and Table 3)

  1. Could the authors explain why they used Sylhet as the geographic area to compare with the rest of the geographic areas.

Response: Thank you. We have made the hange accordingly (Page: 2; Line: 70-71) &  (Page: 4; Line: 152-158).

  1. Throughout the text, the authors report values ​​of p=0. This is an error that should be corrected; if the p-value is very close to 0, they should indicate p<0.001 since there are no p values ​​equal to 0.

Response: Thank you. We have revised accordingly (Table 2 & 3)

  1. For the estimation of the adjusted ORs, in the final step, the independent variables were added to the multiple regression model using the forward stepwise selection method. However, the forward stepwise selection method is used to select the most suitable model, so there may be variables that are not included in the final model. However, as shown by the authors, all the variables were selected despite not contributing to the model. This seems to be wrong. In addition, they do not indicate which were the criteria were used for the selection of these variables in the step-by-step selection method; that is, if the selection of the variables of the model it used the p values, the Akaike information criterion, the Bayesian information criterion or other criteria.

Response: Thank you for your valuable suggestions. Now the text has been written as “First, simple logistic regression analysis was used to examine the bivariate association. Then multiple logistic regression was used to calculate the adjusted odds ratios (aORs) as strength of association between outcome variable and the relevant independent variables. The variables were included in the multiple regression models based on our bi-variate findings as well as literature review” (Page: 4-5; Line: 140-161).

  1. The values ​​of R2 for the models should be described.

Response: Thank you. We have added and described the model diagnostic (Page: 6; Line: 187-192).

  1. In lines 162-164, the authors make the following statement: "these statuses by comparing with geographical regions, we found that the status in Sylhet region was worse than any other regions (Figure 1 and Figure 2)." However, in the text and in the figure, there is no statistical test or p-value to justify this statement.

Response: Thank you for your good suggestions. Now we have added the p-value in the figure. We have also added in the associated factors section in the text (Figure 1 and Figure 2).

  1. Figure 3 is not the most suitable to represent OR values.

Response: We agree and have prepared a separate table. In this figure, there was a typo. We have also revised the figure (Figure 3 and Table 3).

  1. On lines 210-217, the authors show the results of the interactions. However, further explanation of the findings is required to give readers a better sense of the results.

Response: We thank you for your comment and have revised accordingly (Page: 9; Line: 248-256).

Round 2

Reviewer 4 Report

The authors made all corrections